# Ageing Investigation Using Two-Time-Point Metabolomics Data from KORA and CARLA Studies

**DOI:** 10.3390/metabo9030044

**Published:** 2019-03-05

**Authors:** Choiwai Maggie Chak, Maria Elena Lacruz, Jonathan Adam, Stefan Brandmaier, Marcela Covic, Jialing Huang, Christa Meisinger, Daniel Tiller, Cornelia Prehn, Jerzy Adamski, Ursula Berger, Christian Gieger, Annette Peters, Alexander Kluttig, Rui Wang-Sattler

**Affiliations:** 1Research Unit of Molecular Epidemiology, Helmholtz Zentrum München, 85764 Neuherberg, Germany; c.w.chak@utwente.nl (C.M.C.); jonathan.adam@helmholtz-muenchen.de (J.A.); stefan.brandmaier@helmholtz-muenchen.de (S.B.); marcella.covic@gmail.com (M.C.); jialing.huang@helmholtz-muenchen.de (J.H.); christian.gieger@helmholtz-muenchen.de (C.G.); 2Institute of Epidemiology, Helmholtz Zentrum München, 85764 Neuherberg, Germany; peters@helmholtz-muenchen.de; 3Institute of Medical Epidemiology, Biometry and Informatics, Martin-Luther University Halle-Wittenberg, 06108 Halle, Germany; elena.lacruz@uk-halle.de (M.E.L.); daniel.tiller@uk-halle.de (D.T.); alexander.kluttig@uk-halle.de (A.K.); 4German Center for Diabetes Research (DZD), 85764 Neuherberg, Germany; adamski@helmholtz-muenchen.de; 5Independent Research Group Clinical Epidemiology, Helmholtz Zentrum München, 85764 Neuherberg, Germany; christa.meisinger@helmholtz-muenchen.de; 6Research Unit of Molecular Endocrinology and Metabolism, Helmholtz Zentrum München, 85764 Neuherberg, Germany; prehn@helmholtz-muenchen.de; 7Department of Biochemistry, Yong Loo Lin School of Medicine, National University of Singapore, Singapore 117593, Singapore; 8Institute for Medical Informatics, Biometrics and Epidemiology, Ludwig-Maximilians-Universität München, 81377 München, Germany; berger@ibe.med.uni-muenchen.de

**Keywords:** ageing, chronological age, targeted metabolomics, longitudinal study, amino acids

## Abstract

Ageing, one of the largest risk factors for many complex diseases, is highly interconnected to metabolic processes. Investigating the changes in metabolite concentration during ageing among healthy individuals offers us unique insights to healthy ageing. We aim to identify ageing-associated metabolites that are independent from chronological age to deepen our understanding of the long-term changes in metabolites upon ageing. Sex-stratified longitudinal analyses were performed using fasting serum samples of 590 healthy KORA individuals (317 women and 273 men) who participated in both baseline (KORA S4) and seven-year follow-up (KORA F4) studies. Replication was conducted using serum samples of 386 healthy CARLA participants (195 women and 191 men) in both baseline (CARLA-0) and four-year follow-up (CARLA-1) studies. Generalized estimation equation models were performed on each metabolite to identify ageing-associated metabolites after adjusting for baseline chronological age, body mass index, physical activity, smoking status, alcohol intake and systolic blood pressure. Literature researches were conducted to understand their biochemical relevance. Out of 122 metabolites analysed, we identified and replicated five (C18, arginine, ornithine, serine and tyrosine) and four (arginine, ornithine, PC aa C36:3 and PC ae C40:5) significant metabolites in women and men respectively. Arginine decreased, while ornithine increased in both sexes. These metabolites are involved in several ageing processes: apoptosis, mitochondrial dysfunction, inflammation, lipid metabolism, autophagy and oxidative stress resistance. The study reveals several significant ageing-associated metabolite changes with two-time-point measurements on healthy individuals. Larger studies are required to confirm our findings.

## 1. Introduction 

Ageing, the deterioration of functional capacity and stress resistance over a lifetime [1,2], is a main driver of various complex pathologies (e.g., cancer, cardiovascular diseases, metabolic diseases and neurodegeneration) [3,4]. Unlike chronological ageing, which represents the universally constant increase in age in terms of years [3], biological ageing could be influenced by hereditary and environmental factors (e.g., lifestyles and social-economic factors) [5], resulting in great variations in biological ageing rates and health risks among individuals of the same chronological age [6,7]. Hence, developing biomarkers that distinguish between normal and pathological biological ageing process helps us predict the long-term risks of morbidity and mortality [8] and explain individual variations in lifespan [9], which is crucial for achieving healthy ageing.

However, due to the high metabolic heterogeneity of individuals, the development of ageing biomarker remains a great challenge at technical level [1,10,11]. Meanwhile, several studies have shown that many proposed biological ageing pathways (e.g., oxidative stress, inflammatory responses and apoptosis) are intimately associated with metabolic processes. Accordingly, metabolites have been proposed to be the ideal ageing biomarkers—not only are they the most proximal reporters of body alterations due to changes in biological processes [12] but also satisfy all characteristics of an ideal ageing biomarker proposed by Sharman and Zhumadilov [13], they are: i. informative and reflective about the metabolic systems and functional conditions of the body [14]; ii. quantitatively correlated with age [11]; iii. of high reproducibility, sensitivity and specificity [15]; and iv. suitable for use in humans.

Targeted metabolomics, which is a data-driven approach to analyse biochemically known and annotated metabolites in a living organism [6,16,17], captures both short-term and long-term regulation of metabolism as a result of the interactions between nature and nurture [14] and provides objective information on precise metabolic pathways and processes. Via quantitative measurement of the significant metabolite changes over time, not only could it reveal potential ageing biomarkers but also give us insights to the underlying molecular mechanisms [18]— this facilitates our understanding of the changes we experience when we age. Thus, metabolomics has become an upcoming and promising approach to identify candidate age biomarkers [9].

Consequently, previous studies have used metabolomics to explain metabolic changes associated with age in humans [11,15,19,20,21,22] and to identify potential candidate age biomarkers [11,23,24]. However, they were limited by the cross-sectional study designs, where an individual’s real ageing phenotype could not be captured. Furthermore, the results could be confounded by intra-personal (e.g. metabolic dynamics) and inter-personal variances (e.g., lifestyles, environmental and genetic factors) [2,6]. This leaves us a question: are there any metabolites that commonly change in our body when we age? In response to this question, a large population-based longitudinal analysis with at least two separate time-points is necessary to track the changes in metabolite concentration of each individual over time [6,9,10]. Correspondingly, the overall metabolite concentration changes during the ageing process within a healthy population could be observed. However, to our knowledge, such a population-based longitudinal analysis is by far limited in targeted metabolomics studies regarding to human ageing [6]. Herein, this study aims to fill the knowledge gap.

The primary objective of the study is to identify metabolites that commonly change among healthy individuals over time. The secondary objective is to understand the potential biochemical roles of the identified metabolites in ageing.

## 2. Results

### 2.1. Characteristics for the Discovery and Replication Population

In KORA, a total of 317 healthy women and 273 healthy men were included for analysis (Figure 1A). Replication was conducted using independent sample of 195 women and 191 men from CARLA study (Figure 1B).

The characteristics of study participants at two time-points for KORA and CARLA studies are shown in Table 1. Both study participants were of the same age range (55–74 years old). Both women and men were of comparable mean chronological age at baseline in both discovery and replication studies (around 63 years old). In general, women showed healthier lifestyles (more physically active, less smokers and lower alcohol intake) when compared with men in both studies (Table 1). For example, in KORA baseline study, 52.7% women and 47% men were physically active, whereas seven years later, 58.7% women and 55.3% men were active, respectively. Similar trends were observed in the CARLA study (42.6% women and 33.5% men were physically active at baseline, while 53.9% women and 42.9% men were active during the four years follow-up). Overall, participants during the follow-ups became more physically active (e.g., 6% women and 8.3% men in KORA and 11.3% women and 9.4% men in replication, respectively). Although there were significant increases in mean BMI (around 0.3 to 0.4 $\frac{\mathrm{kg}}{m^{2}}$) in women in both studies and men (0.2 $\frac{\mathrm{kg}}{m^{2}}$) in CARLA, significant decreases in mean systolic blood pressure for both sexes were observed in both studies (around 4–5 mmHg).

### 2.2. Identification of Ageing-Associated Metabolite

Out of the 122 analysed metabolites in KORA (Appendix A), 72 and 81 metabolites were significantly (*p* < 4.1 × 10^−4^) associated with ageing in women and men, respectively (Appendix A). These associations are independent from chronological age at baseline, BMI, physical activity, smoking status, alcohol intake and SB pressure. The serum concentration of more than a half of analysed lipids (acylcarnitines, diacyl-phosphatidylcholines (PCaa), acyl-alkyl-PCs (PCae)) and amino acids were significantly associated with ageing. Furthermore, we found that all analysed lysoPCs and all sphingomyelins were negatively associated with ageing (Appendix A). Out of the 85 ageing-associated metabolites, 68 metabolites (about 80%) were observed in both sexes in the KORA study, whereas 17 metabolites were found to be sex-specific (Appendix A). 

Among the 85 metabolites identified in KORA, 60 metabolites were available in CARLA dataset for replication (53 for women and 58 for men, Appendix A). Based on false discovery rate-adjusted P values (pFDR), a total of five and four metabolites were replicated in women and men, respectively (Table 2). Significantly decreased concentration of arginine and increased ornithine were identified in both sexes. Increased levels of octadecanoylcarnitine (C18), serine and tyrosine were replicated in women whereas decreased concentration of PC aa C36:3 and PC ae C40:5 were replicated in men (Figure 2, Table 2). 

The replicated metabolite with the greatest change over time (in terms of year) was ornithine, with a yearly average increase by 0.14 standard deviations in KORA in women and men, respectively and by 0.24 and 0.22 in CARLA in women and men, respectively. 

In addition to the replicated metabolites mentioned above, additional five metabolites (namely octadecenoylcarnitine (C18:1), phenylalanine, valine, lysoPC a C16:0 and lysoPC a C18:0) were significant in the CARLA study but in opposite trends between discovery and replication studies: phenylalanine (women: ß = −0.07 in KORA; ß = 0.07 in CARLA; men: ß = −0.09 in KORA; ß = 0.07 in CARLA) (Appendix A).

### 2.3. Ageing Related Pathways

Based on literature search and the Kyoto Encyclopaedia of Genes and Genomes (KEGG) database, we found that three amino acids, ornithine, serine and arginine, are involved in different pathways related to the ageing processes. Their metabolic pathways and/or cell-signalling functions are shown in Figure 3. 

## 3. Discussion

Ageing is difficult to be captured due to high individual variances along time. Limited by the study design, previous large-scale, cross-sectional metabolomics studies of ageing [11,23] were unable to address two major issues of ageing biomarkers: (1) capturing the true “ageing” phenotype, rather than “chronological age”; and (2) taking the metabolic heterogeneity of individuals into account. These limitations could only be overcome by analysing individual metabolic profiles with at least two-time-point measurements [10]. 

Hence, with the availability of two-time-point measurements, our study has made the capture of the “ageing” phenotype with consideration of individual heterogeneity in ageing possible. By pooling the individual data of metabolite changes over time after adjusting for potential confounders that may affect ageing rate and metabolite concentration, namely chronological age at baseline, BMI and lifestyle factors, what we observe is, consequently, the average of changes in metabolite concentration within a healthy population during ageing, irrespective of individual differences. Therefore, the metabolites we identified are associated with ageing process in general.

In both KORA and CARLA studies, we observed similar degree of change in replicated metabolites. However, it was noteworthy that a number of metabolites (namely C18:1, phenylalanine, valine, lysoPC a C16:0 and lysoPC a C18:0) demonstrated significant but opposite changes between KORA and CARLA participants. There are some potential factors that may account for the differences, for instance, the concentrations of these metabolites are systematically different between fasting (in KORA) and non-fasting samples (in CARLA) because they could be influenced by the time since the last food intake and activities [25,26]. Furthermore, these metabolites may not change linearly (e.g., but parabolically) over time, as the changes in different time span (seven years in KORA and four years in CARLA) were captured. In fact, a number of metabolites including phenylalanine and valine were reported to be influenced by food intake [27]. 

Consistent with previous studies that suggested metabolic profiles are highly sex-dependent [11,15,28,29], we also observed distinctive metabolite signatures between women and men during ageing. For instance, sex-specific changes are observed in the metabolites commonly changed in both sexes over time (e.g. C18 and PC ae C40:5) where both sexes demonstrate different peaks in baseline and follow-up studies, indicating their changes are not only time-dependent but also sex-dependent. Ageing-related decrease in several PC ae metabolites (also known as choline plasmalogens) observed and replicated in ageing men in our study was previously reported in a study of elderly subjects compared with young controls [30]. Such sex differences might be attributed to the differences in the roles of sex chromosomes, energy metabolism, adipose tissue storage and glucose homeostasis [31] or sex hormone changes in postmenopausal women [32], leading to large changes in fatty acid beta-oxidation and phospholipid, hormone and amino acid metabolism upon ageing [33]. Therefore, our results again highlight the importance of sex stratification in studying ageing with metabolite concentration profiles. 

Furthermore, it is also noteworthy that serine, which was discovered and replicated in women only, also commonly increased in both sexes over time and was significant in men under FDR-adjusted P value in discovery and replication studies. This indicates that despite its insignificant increase in women over time, this metabolite could also be associated with ageing in general. However, our study power to detect the changes was potentially limited by the sample size.

### 3.1. Potential Roles of Identified Metabolites in Ageing

Arginine and ornithine are involved in the metabolism of polyamines (putrescine, spermidine, spermine) whose supplementation in several model organisms led to increased lifespans [34]. Decreased arginine levels we found in ageing participants of KORA and CARLA could be caused by its increased metabolism through arginase and nitric oxide synthase. Increased expression of arginase was found in vascular models of ageing [35] and its inhibition was suggested to reduce systemic and vascular inflammation [36]. In addition to significantly decreased arginine levels, we also replicated significant increase in ornithine upon ageing. Ornithine is decarboxylated by the enzyme ornithine decarboxylase (ODC) to produce polyamines (putrescine, spermidine and spermine). Enzymatic activities involved in polyamine synthesis (e.g., ODC and spermidine synthase) are known to decrease with ageing, resulting in lower polyamine production with increasing age [34]. Decreased activity of ODC and lower synthesis of polyamines was also reported to contribute to ageing-associated decline in oocyte maturation [37]. Spermidine, one of the polyamines, has several anti-ageing properties such as increased lifespan, improved stress resistance and reduced age- associated pathologies via increased autophagy, lower inflammation and regulation of lipid metabolism and cell growth [34]. 

Whether ageing-associated increase of the C18 in women and decrease of PC aa C36:3 and PC ae 40:5 in men of the KORA and CARLA studies could be related to low polyamines or developing pre-diabetes [38] remain to be further investigated. Yet, since the trends of other acylcarnitines and phosphatidylcholines in relation to biological ageing were inconsistent (both increases and decreases were observed) in KORA and were not replicated in CARLA (the latter probably due to non-fasting status), further studies will be needed to understand their potential links with ageing. In summary, risen ornithine observed in our ageing participants might support the spermidine hypothesis of ageing and could be linked with altered lipid metabolism. 

Ageing KORA and CARLA individuals also demonstrated increased levels of serine, a non-essential amino acid and a predominant source of one carbon groups for the de novo synthesis of purine nucleotides that is essential for cell proliferation. Serine has been recently linked with decreased lifespan in yeast models through sensitization to oxidative stress and activation of the TOR-S6 signalling pathway [39].

Meanwhile, ageing-related increase of tyrosine in KORA and CARLA women could be due to alterations in the activity of tyrosine hydroxylase (TH) and synthesis of catecholamines such as dopamine and other neurotransmitters involved in stress response. In mouse models, a haplo-insufficiency in TH was beneficial for regulation of blood pressure and increased lifespan [40].

### 3.2. Strengths and Limitations

This study adds to the relatively less investigated field of human ageing by metabolomics using a longitudinal study design. It has the privilege of making use of the longitudinal targeted metabolomics data of healthy individuals from two independent human cohort studies. It also fills the current knowledge gap by capturing the change in metabolites during ageing by making the observation of the individual changes in metabolite concentration over time possible and accounting for the intra- and inter-personal variances, such as BMI and lifestyle changes over time. Furthermore, the similarity of participant characteristics in both studies has made our results generally comparable. Moreover, both KORA and CARLA studies made use of serum samples, which are generally more subject to homeostatic control and hence generally less sensitive to intervention-induced metabolic phenotype changes compared to urine samples. Despite urine samples have an advantage of non–invasive sampling method, appropriate normalization methods for the metabolite concentration profiles should be considered when adopted [41,42]. Future studies should consider performing longitudinal analyses with urine samples, so as to compare the results with serum samples. 

Another major strength of this study is the replication from another independent CARLA study. After taken into account the potential influences of inter-kit differences on metabolite concentration in KORA study, CARLA study serves as an excellent replication study, since the metabolites in CARLA were measured with the same assay at both time-points. Hence, the fact that several amino acids were successfully replicated with similar degree of changes over time further adds robustness to the evidence in the study. 

Nevertheless, our study also exhibits several limitations. Firstly, the difference between fasting and non-fasting samples may influence the concentration level of certain metabolites, especially those from the small lipid class. Secondly, since only two time-points were available, it provided limited information on whether individuals experience consistent changes in metabolite concentration along time. Thirdly, the generalizability of the results was limited to the German population where individuals may share similar lifestyle factors and genetic backgrounds. Fourthly, given the complexity of ageing process, future studies should consider performing systems analyses on multi-OMICs data to examine multiple biomarkers apart from metabolomics. Finally, it is noteworthy that despite some sex-specific metabolite changes, many of them also displayed changes in the same direction in both sexes, even though they were borderline significant. Our literature review of published studies pointed to a very low reproducibility of ageing-related metabolites, especially those in lipid class. Thus, future studies with larger sample size and multi-time-point data from other fasting populations are highly recommended to confirm our results.

## 4. Materials and Methods

### 4.1. Ethics Statement

All participants in KORA and CARLA studies gave written informed consent. The KORA study was approved by the local ethics committee of the Bavarian Medical Association, Germany. The project identification codes for KORA S4 and KORA F4 are 99186 and 06068, respectively. The CARLA study was approved by the local ethics commission of the Medical Faculty of the Martin Luther University of Halle-Wittenberg and was carried out in accordance with the declaration of Helsinki. The ethic codes for CARLA-0 and CARLA-1 are 117/24.10.01/11 and 164/12.10.05/1, respectively. 4.2. KORA Studies 

The “Kooperative Gesundheitsforschung in der Region Augsburg” (KORA) Study is a large population-based cohort conducted in Augsburg region, Germany [11]. It consists of interviews, medical and laboratory examinations and biological sample collection [11]. In the baseline study (KORA S4) (1999–2001), a total of 4,261 individuals were examined, while a total of 3,080 participants took part in the seven-year follow-up study (KORA F4) (2006–2008) [43,44].

### 4.2. CARLA Studies 

The CARLA-Study (Cardiovascular Disease, Living and Ageing in Halle) is a population-based cohort study in an elderly population of the city Halle/Saale in eastern Germany [45]. Study design and methods were described in detail elsewhere [45]. For the baseline study (CARLA-0), 1,779 participants (46% women) aged between 45 and 83 years old were examined between July 2002 and January 2006. The first four-year (SD = 0.3) follow-up examination (CARLA-1) with 1,436 participants (45% women) was conducted between March 2007 and March 2010.

### 4.3. Sample Collection

In KORA, blood was drawn into S-Monovette® serum gel tubes (SARSTEDT AG & Co., Nümbrecht, Germany) in the morning between 8:00 a.m. and 10:30 a.m. after a fasting period of at least eight hours [17]. Blood samples were processed according to standard procedures. The samples were stored at −195.79 °C in liquid nitrogen until the execution of metabolic analyses. Details were reported in previous publications [11,17].

In CARLA, metabolic profiles in the blood serum of the study participants were measured. Blood samples were taken after a supine rest of 30 min. Followed by a 10 min centrifugation (20 °C, 1500 revolutions per min), the samples was collected and deep frozen to −80 °C on the same day until analysis of the metabolites [45].

### 4.4. Metabolite Quantification 

Targeted metabolomics measurements have been performed in the Metabolomics Platform of the Helmholtz Zentrum München. In KORA, serum samples from participants at the baseline (KORA S4) and follow-up (KORA F4) studies were measured with the Absolute*IDQ*^TM^ p180 Kit and Absolute*IDQ*^TM^ p150 Kit (Biocrates Life Sciences AG, Innsbruck, Austria), respectively [17,28]. In CARLA, serum samples were measured by Absolute*IDQ*^TM^ p150 Kit [46]. Serum samples have been analysed using flow injection-electrospray ionization-tandem mass spectrometry (FIA-ESI-MS/MS) and, in the case of p180 assay, additionally using liquid chromatography-electrospray ionization-tandem mass spectrometry (LC-ESI-MS/MS). Details of assay procedures of both assays as well as nomenclature were previously reported [47,48].

Quality control (QC) procedures were performed for both studies and details were reported elsewhere [11,17]. In KORA, 188 and 163 metabolites were measured with Absolute*IDQ*^TM^ p180 Kit (KORA S4) and Absolute*IDQ*^TM^ p150 Kit (KORA F4). A detailed QC for Absolute*IDQ*^TM^ p180 Kit was reported previously in Appendix A for characteristics of the 188 targeted metabolites [17]. A total of 123 metabolites were measured in both time-points, including sum of hexoses (H1), 20 acylcarnitines, 14 amino acids, 13 sphingomyelins, diacyl and acyl-alkyl phosphatidylcholines (33 PC aas and 34 PC aes) and eight lysophosphatidylcholines (LysoPCs) (Appendix A). In CARLA, 134 metabolites passed quality control [45]. 

### 4.5. Pre-Processing Of Metabolite Data

Since different assays (Absolute*IDQ*^TM^ p180 Kit and Absolute*IDQ*^TM^ p150 Kit) were used for quantitative measurements of metabolites in KORA S4 and KORA F4, respectively, inter-kit normalization was performed. For both assays, the same reference samples (human plasma pooled material, Seralab) were measured five times on each kit plate. QC procedures were performed on these reference samples. In total, 114 and 351 reference samples were initially analysed for Absolute*IDQ*^TM^ p180 Kit and Absolute*IDQ*^TM^ p150 Kit respectively. Sample values that exceed the range of median ±1.5 inter-quartile range were identified as outlier values. Samples which contained more than or equal to 10 outlier values were defined as outlier samples and were excluded (*N* = 3 in Absolute*IDQ*^TM^ p180 Kit; *N* = 23 in Absolute*IDQ*^TM^ p150 Kit). After exclusion of outlier samples, coefficient of variation (CV) of each metabolite in each kit were calculated separately using the remaining 111 and 328 samples respectively. Metabolites which had a CV larger than 25% in either KORA S4 or KORA F4 were excluded from the analysis. Accordingly, one metabolite (C9), which had a CV value of 36.96% (>25%) in KORA F4 (refer to Appendix A) was excluded, resulting in a total of 122 metabolites eligible for normalization. 

Afterward, the results of the remaining reference samples were used for normalization of the 122 metabolites. The normalization factor (NF) for each metabolite were calculated by dividing the median of reference sample values in Absolute*IDQ*^TM^ p180 Kit (used in KORA S4) by the median of reference sample values in Absolute*IDQ*^TM^ p150 Kit (used in KORA F4). Afterwards, the metabolite concentration of each sample in KORA F4 was adjusted by multiplying the concentration values to the NF of their corresponding metabolites (Appendix A). Median values report by existing literature [26,29] were included for comparison (Appendix A).

Since some metabolites had skewed distribution, all metabolite concentrations were natural-log transformed and scaled (z-transformed) for a more symmetric distribution before statistical analyses. 

### 4.6. Inclusion and Exclusion Criteria of Study Samples

In KORA, a total of 1125 individuals whose serum samples were collected at both baseline and follow-up studies were included for analysis. Since diets have notable influences on metabolite concentration and may lead to confounding results, we excluded 126 individuals with non-fasting samples at either time point (Figure 1A). In order to study the influence of normal ageing process on metabolic profiles in generally healthy population, only healthy individuals without major metabolic diseases were included. Therefore, we excluded 277 individuals with cardiovascular infarction at either or both baseline and follow-up studies, as well as diabetes (both type I and II diabetes, where disease status was identified based on physician-validated and self-reported diagnoses, fasting glucose and two-hour post glucose load and information on medication), 87 individuals with hypertension (SB pressure > 160 mmHg) and 56 individuals with obesity (BMI > 35 $\frac{\mathrm{kg}}{m^{2}}$). As a result, a total of 590 individuals were included (Figure 1A).

In CARLA, the same criteria were applied except non-fasting samples were used as majority of CARLA samples were non-fasted. Among the 1409 individuals whose blood samples were available in both baseline and follow-up studies, 395 individuals with cardiovascular disease or diabetes, 241 individuals with hypertension and 50 individuals with obesity in either or both baseline and follow-up were excluded (Figure 1B). In addition, 337 individuals with baseline chronological age below 55 or above 74 years were excluded for better comparability to the KORA studies after accounting for potential birth cohort difference. As a result, 386 participants were included.

### 4.7. Statistical Analysis

Owing to the significant metabolic [15,28] and biological ageing differences between sexes [49,50] reported in previous studies, sex-stratified analyses were performed to analyse the sex-specific metabolite changes among individuals at two time-points. 

Generalized estimation equation (GEE) model was performed on each metabolite (assuming the correlation structure as equally correlated within subjects over time) to identify any significant changes in metabolite concentration during normal ageing process in terms of years, after adjusted for the intra- and inter-personal confounders within two time-points. Baseline chronological age [11], BMI [11] and potential lifestyle confounders such as physical activity, smoking status, alcohol intake and systolic blood pressure [5], which were known to be the significant confounders affecting metabolite concentration over time from previous literature, were included into the model:

Metabolite | id = ß0 + ß1*(year) + ß2*(chronological age at baseline) + ß3*(BMI) + ß4*(physical activity) + ß5*(smoking status) + ß6*(alcohol intake) + ß7*(SB pressure) + ε.

The outcome variable of the model was the concentration of each metabolite, while the major predictor variable of interest was the beta estimate of year, which represents the overall changes in metabolite concentration each year. 

To identify significantly changed metabolites in both women and men, in the discovery study (KORA), we adopted stringent Bonferroni correction for determining the cut-off of significance level. 

After identifying the significantly changed metabolites based on stringent Bonferroni correction, to further determine whether a metabolite is replicated in the replication study (CARLA), less stringent false discovery rate-adjusted P value (pFDR) will be adopted, given the smaller sample size in CARLA study and the discovery nature in this paper. Nevertheless, both significant P values after Bonferroni correction and FDR adjustment were displayed and indicated for reference.

All analyses were performed and the graphs were generated using the R programming language version 3.1.3 (http://www.r-project.org/) (R Development Core Team, Vienna, Austria). Packages used were broom (David Robinson, 2016), tableone (Kazuki Yoshida & Justin Bohn, 2015), joineR (Graeme L. Hickey, 2017), gee (Vincent J Carey, Thomas Lumley & Brian Ripley, 2015), gdata (Gregory R. Warnes, 2017), scales (Hadley Wickham, 2016) and graphics (R Development Core Team, 2017).

### 4.8. Pathway Analysis

Potential metabolic pathways related to the identified metabolites were investigated using literature search and a web database for pathway analysis: KEGG Pathways (http://www.genome.jp/kegg/pathway.html). 

## 5. Conclusions

The study reveals several significant ageing-associated metabolite changes with two-time-point measurements on the same individuals in two German populations. For instance, the decrease in arginine and the increase in ornithine and serine over time may be involved in ageing processes: apoptosis, mitochondrial dysfunction, inflammation, lipid metabolism, autophagy and oxidative stress resistance. Future studies with larger sample size and multi-time-point data should be conducted to confirm our findings.

## Figures and Tables

**Figure 1 metabolites-09-00044-f001:**
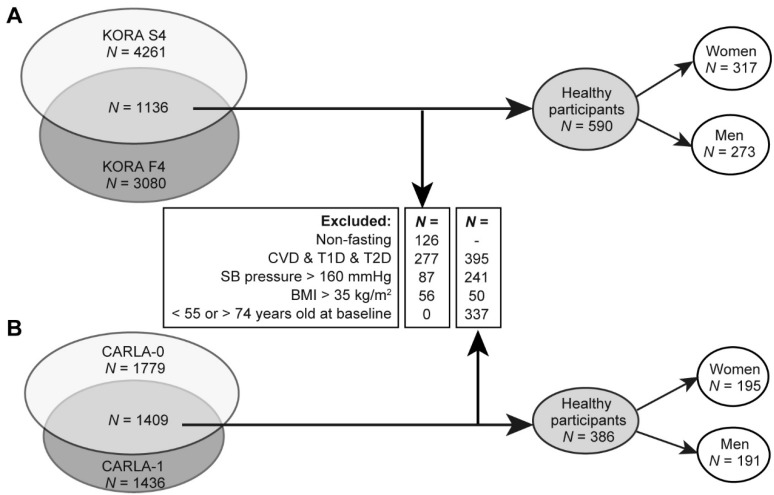
Flow diagram showing the inclusion and exclusion procedures of the study population in KORA (**A**) and in CARLA (**B**). CVD, cardiovascular disease. T1D, type 1 diabetes. T2D, type 2 diabetes. SB, systolic blood. BMI, body mass index.

**Figure 2 metabolites-09-00044-f002:**
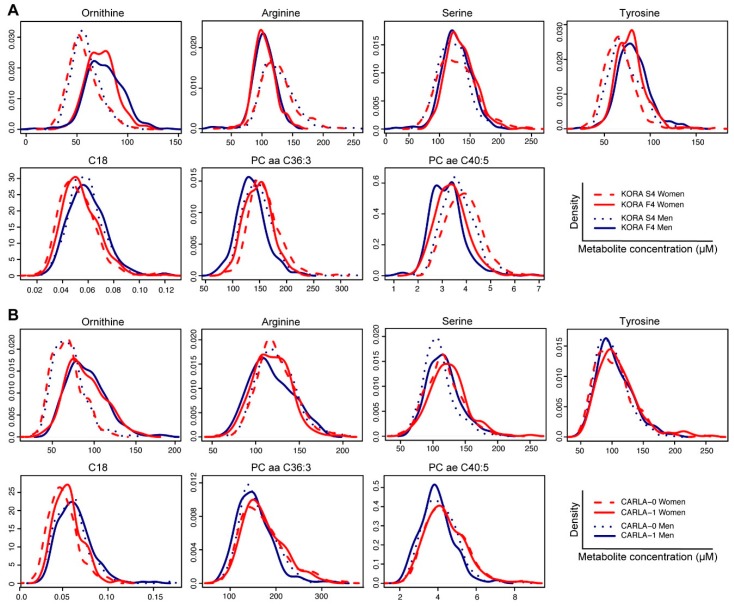
Density plot showing the distribution of metabolite concentration of both women and men in two time-points in KORA S4 and F4 (plot A) and in CARLA-0 and CARLA-1 (plot B). C18, octadecanoylcarnitine. PC aa, diacyl phosphatidylcholine. PC ae, acyl-alkyl phosphatidylcholine.

**Figure 3 metabolites-09-00044-f003:**
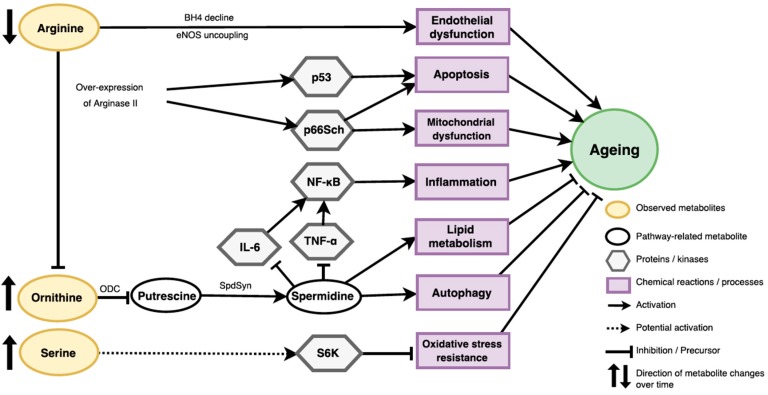
Potential metabolic pathways of arginine, ornithine and serine related to biological ageing. BH4, tetrahydrobiopterin. eNOS, endothelial nitric oxide synthases. p53, p53 kinase. p66Sch, p66Sch kinase. NF-κB, complex nuclear factor kappa B. IL-6, interleukin 6. TNF-α, tumour necrosis factor alpha. ODC, ornithine decarboxylase. SpdSyn, spermidine synthase. S6K, S6 kinase.

**Table 1 metabolites-09-00044-t001:** Characteristics of study participants at baseline and follow-up in women from KORA and CARLA (stratified by time-point). BMI, body mass index. SB, systolic blood. SD, standard deviation.

Women							Men					
	Women in Discovery KORA (*N* = 317)			Women in Replication CARLA (*N* = 195)			Men in Discovery KORA (*N* = 273)			Men in Replication CARLA (*N* = 191)		
Variables	KORA S4 (Baseline)	KORA F4 (Follow-up)	*p*-Value ^a^	CARLA-0 (Baseline)	CARLA-1 (Follow-up)	*p*-Value ^a^	KORA S4 (Baseline)	KORA F4 (Follow-up)	*p*-Value ^a^	CARLA-0 (Baseline)	CARLA-1 (Follow-up)	*p*-Value ^a^
Chronological age, years (mean (range))	62.71 (55–74)	69.71 (62–81)	-	63.50 (55–74)	67.51 (59–78)	-	62.69 (55–74)	69.69 (62–81)	-	63.40 (55–74)	67.41 (59–79)	-
$\mathrm{BMI},\frac{\mathrm{kg}}{m^{2}}$ (mean (SD))	26.88 (3.41)	27.21 (3.53)	7.0 × 10^−4^	26.87 (3.34)	27.24 (3.43)	9.6 × 10^−5^	27.36 (2.82)	27.49 (3.00)	0.115	27.31 (3.08)	27.51 (3.26)	0.01
Physically active (%) ^b^	52.7	58.7	0.0402	42.6	53.9	0.003	47.0	55.3	0.017	33.5	42.9	0.009
Non-smokers (%)	89.9	93.4	0.0055	84.6	86.7	0.1573	82.7	89.7	1.0 × 10^−4^	83.2	86.4	0.01
Low alcohol intake (%) ^c^	86.8	89.0	0.3711	94.4	94.4	0.99	77.4	82.8	0.037	88.0	92.2	0.05
SB pressure, mmHg (mean (SD))	125.54 (15.94)	121.38 (16.37)	3.2 × 10^−5^	132.34 (14.76)	129.99 (13.38)	0.03	131.79 (14.53)	127.58 (14.93)	2.3 × 10^−5^	138.99 (12.65)	133.14 (13.93)	1.4 × 10^−11^

^a^ To compare the differences between two time-points, paired *t*-tests were performed for continuous variables. McNemar’s tests were performed for binary variables. Significant *p*-values at 5% level were highlighted in bold. ^b^ More than one hour of sports per week in at least one of the summer or winter seasons. ^c^ Daily alcohol intake ≤ 20 g in women and ≤ 40 g in men.

**Table 2 metabolites-09-00044-t002:** Replicated changes in metabolite concentration in women and men from KORA and CARLA studies. This table shows the replicated changes in metabolite concentration among women and men after false discovery rate adjustment at 5% level. Beta estimates (ß) and confidence intervals (CI) were calculated using the multivariate generalized estimation equation (GEE) model. The model was adjusted for chronological age at baseline, body mass index, physical activity, smoking status, alcohol intake and systolic blood pressure. Significant *p* values after Bonferroni correction (cut-offs: *p*-value < $\frac{0.05}{122}$ = 4.1 × 10^−4^ in discovery; *p*-value < $\frac{0.05}{53}$ = 9.5 × 10^−4^ in women and *p*-value < $\frac{0.05}{58}$ = 8.7 × 10^−4^ in men in replication) and false discovery rate-adjusted *p*-values (pFDR) at 5% level were highlighted in bold. C18, octadecanoylcarnitine. PC aa, diacyl phosphatidylcholine. PC ae, acyl-alkyl phosphatidylcholine.

**Metabolites**	**Women in Discovery KORA (*N* = 317)**			**Women in Replication CARLA (*N* = 195)**		
	**ß (95% CI)**	***p*-Value**	**pFDR**	**ß (95% CI)**	***p*-Value**	**pFDR**
Ornithine	0.14 (0.13, 0.16)	**4.6 × 10^−82^**	**1.1 × 10^−80^**	0.24 (0.21, 0.27)	**6.9 × 10^−48^**	**4.0 × 10^−46^**
Arginine	−0.10 (−0.12, −0.09)	**3.2 × 10^−30^**	**1.5 × 10^−29^**	−0.06 (−0.10, −0.03)	1.1 × 10^−3^	**1.0 × 10^−2^**
Serine	0.04 (0.02, 0.06)	**3.0** **× 10^−4^**	**5.1** **× 10^−4^**	0.01 (0.01, 0.02)	1.1 × 10^−3^	**1.1 × 10^−2^**
Tyrosine	0.11 (0.09, 0.13)	**1.6 × 10^−36^**	**9.2 × 10^−36^**	0.05 (0.01, 0.08)	6.8 × 10^−3^	**4.9 × 10^−2^**
C18	0.03 (0.01, 0.05)	**3.7 × 10^−4^**	**6.2 × 10^−4^**	0.03 (0.02, 0.04)	**6.1 × 10^−8^**	**1.2 × 10^−6^**
**Metabolites**	**Men in Discovery KORA (*N* = 273)**			**Men in Replication CARLA (*N* = 191)**		
	**ß (95% CI)**	***p*-value**	**pFDR**	**ß (95% CI)**	***p*-value**	**pFDR**
Ornithine	0.14 (0.12, 0.16)	**2.7 × 10^−56^**	**4.2 × 10^−55^**	0.22 (0.19, 0.25)	**2.8 × 10^−41^**	**1.6 × 10^−39^**
Arginine	−0.12 (−0.14, −0.09)	**5.1 × 10^−29^**	**2.4 × 10^−28^**	−0.07 (−0.10, −0.03)	**7.0 × 10^−5^**	**1.4 × 10^−3^**
PC aa C36:3	−0.05 (−0.07, −0.03)	**2.2 × 10^−8^**	**4.9 × 10^−8^**	−0.05 (−0.08, −0.02)	2.3 × 10^−3^	**2.1 × 10^−2^**
PC ae C40:5	−0.09 (−0.11, −0.08)	**1.2 × 10^−31^**	**5.6 × 10^−31^**	−0.06 (−0.09, −0.03)	5.4 × 10^−4^	**6.2 × 10^−3^**

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
