# Peer review of "Ageing Investigation Using Two-Time-Point Metabolomics Data from KORA and CARLA Studies"

_metabolites, 2019, doi:10.3390/metabo9030044_

Reviewer 1 Report

Review

Manuscript ID: metabolites-417081

Title: Ageing investigation using two-time-point metabolomics data from KORA and CARLA studies

Authors: Choiwai Maggie Chak, Maria Elena Lacruz, Jonathan Adam, Stefan Brandmaier, Marcela Covic, Jialing Huang, Christa Meisinger, Daniel Tiller, Cornelia Prehn, Jerzy Adamski, Ursula Berger, Christian Gieger, Annette Peters, Alexander Kluttig, Rui Wang-Sattler

A recent biomarker study of aging came under some heavy criticism in a letter to PNAS.  The letter draws attention to four key issues in any aging investigation:

(i)                  metabolic heterogeneity of individuals,

(ii)                cross-sectional vs. longitudinal study design,

(iii)               multiple statistical tests, and

(iv)               independent biological replication.

The present work represents an effort to address these issues by combining results from two different studies, KORA and CARLA. Trabado et al. [PLOS ONE DOI:10.1371/journal.pone.0173615] have outlined that KORA recruitment was performed “in the general population, lacking a complete and precise clinical and biological healthy status”. However, we consider this is not a major problem, the real problem with this study is the way the data analysis is presented.

All data is reported in a ‘statistical’ way without reference values (mean and median, µmol/L, for the amino acids and other metabolites), so there is no way to compare it with data from previous studies. These comparisons are important because most metabolites reported in this study are present in many interconnected pathways and are influenced by genetic inheritance, exercise and diet. However, using Figure 2 we have approximated some values for the amino acids and we have compared it with the daily variations reported by Thompson et al.  [Metabolomics. 2012; 8(4): 556–565. doi:10.1007/s11306-011-0345-9]. (These authors characterized daily variation of amino acids and acylcarnitines in response to feeding and activity by measuring serum metabolites at various times and after various activities during the day). Unfortunately, the metabolite concentration variations in the present study are not pronounced enough to be considered as potential aging ‘biomarkers’.  Trabado et al. [PLOS ONE DOI:10.1371/journal.pone.0173615] also reported reference values for AAs and ACs that can be used for this comparison too.

In our opinion and based on the manuscript results, the reported metabolites do not satisfy the requirements of specificity and sensitivity to be considered potential biomarkers. Also, to address the ‘aging problem’, we probably need to move beyond simple approaches to a systems analysis approach that examines multiple biomarkers at once [Zierer, J., Menni, C., Kastenmüller, G., & Spector, T. D. (2015). Integration of ‘omics’ data in aging research: from biomarkers to systems biology. Aging cell, 14(6), 933-944.]. Having said that, we think this manuscript can be useful to the community, particularly the idea of combining multiple studies to compensate for some of the above-mentioned issues. I would suggest adding a table with the concentration variations of the most important metabolites and the comparison with similar values from previous studies. For example, compare the concentration fluctuations in the present work to the daily variations reported by Thompson et al.  [Metabolomics. 2012; 8(4): 556–565. doi:10.1007/s11306-011-0345-9].

Author Response

Response to Reviewer 1 Comments

A recent biomarker study of aging came under some heavy criticism in a letter to PNAS.  The letter draws attention to four key issues in any aging investigation:

(i)                  metabolic heterogeneity of individuals,

(ii)                cross-sectional vs. longitudinal study design,

(iii)               multiple statistical tests, and

(iv)               independent biological replication.

The present work represents an effort to address these issues by combining results from two different studies, KORA and CARLA. Trabado et al. [PLOS ONE DOI:10.1371/journal.pone.0173615] have outlined that KORA recruitment was performed “in the general population, lacking a complete and precise clinical and biological healthy status”. However, we consider this is not a major problem, the real problem with this study is the way the data analysis is presented.

All data is reported in a ‘statistical’ way without reference values (mean and median, µmol/L, for the amino acids and other metabolites), so there is no way to compare it with data from previous studies. These comparisons are important because most metabolites reported in this study are present in many interconnected pathways and are influenced by genetic inheritance, exercise and diet. However, using Figure 2 we have approximated some values for the amino acids and we have compared it with the daily variations reported by Thompson et al.  [Metabolomics. 2012; 8(4): 556–565. doi:10.1007/s11306-011-0345-9]. (These authors characterized daily variation of amino acids and acylcarnitines in response to feeding and activity by measuring serum metabolites at various times and after various activities during the day). Unfortunately, the metabolite concentration variations in the present study are not pronounced enough to be considered as potential aging ‘biomarkers’.  Trabado et al. [PLOS ONE DOI:10.1371/journal.pone.0173615] also reported reference values for AAs and ACs that can be used for this comparison too.

In our opinion and based on the manuscript results, the reported metabolites do not satisfy the requirements of specificity and sensitivity to be considered potential biomarkers. Also, to address the ‘aging problem’, we probably need to move beyond simple approaches to a systems analysis approach that examines multiple biomarkers at once [Zierer, J., Menni, C., Kastenmüller, G., & Spector, T. D. (2015). Integration of ‘omics’ data in aging research: from biomarkers to systems biology. Aging cell, 14(6), 933-944.]. Having said that, we think this manuscript can be useful to the community, particularly the idea of combining multiple studies to compensate for some of the above-mentioned issues. I would suggest adding a table with the concentration variations of the most important metabolites and the comparison with similar values from previous studies. For example, compare the concentration fluctuations in the present work to the daily variations reported by Thompson et al.  [Metabolomics. 2012; 8(4): 556–565. doi:10.1007/s11306-011-0345-9].

Response: We thank this reviewer for his willingness to participate in the peer-review processing of our manuscript and her/his constructive comments to improve our study.

We do agree with this reviewer is that ‘KORA is population-based cohort study, lacking a complete and precise clinical and biological healthy status’. Therefore, in our study, we have extensively excluded 535 individuals with cardiovascular infarction at either or both baseline and follow-up studies, as well as diabetes (both type I and II diabetes, where disease status was identified based on physician-validated and self-reported diagnoses, fasting glucose and two-hour post glucose load, and information on medication), individuals with hypertension and with obesity. As a result, a total of 590 individuals were included (as shown in Figure 1A).

We additionally added one Table: supplementary table S4, which shows the mean, standard deviation and median values of the metabolite concentration among KORA participants at two time-points, which are also the values used for Figure 2. On top of that, we have compared the median of reference values of Trabado et al. (2017) and the values reported by Thompson et al. (2012) in supplementary table S1 for comparison.

We do agree that systematic approaches integrating multiple levels of OMICs data with larger samples size and additional time points would enable a deeper understanding of the aging process. However, our study is based on the available currently data. The data within the KORA cohort is owned by multiple instances and individuals, which makes it almost impossible to combine the suggested amount of data in one study.

Reviewer 2 Report

This manuscript presents results from a longitudinal study of the targeted metabolome at two time points in a human longitudinal study cohort, with replication carried out in a second cohort. The authors use Generalized Estimating Equations to control for a variety of health-related traits (e.g., smoking, physical activity, blood pressure, chronological age), and then test for significant effects of time on the level of individual metabolites.

Over half the metabolites in the first (KORA) cohort were associated with what the authors refer to as “biological ageing”, though only a handful of these metabolites were replicated in the second study.

The authors claim that this the first longitudinal study of metabolites in humans. Previous work has already been done in this regard (e.g. Mielke et al. 2014 Aging Cell; Saleem et al. In Press. Journals of Gerontology), though there are indeed relatively few studies, and as such, this work would provide further valuable data regarding our understanding of how metabolite levels change with age.

Although the data are valuable, I have some concerns regarding the interpretation of these results.  The authors claim that this study measures the effect of biological ageing rather than chronological ageing. One needs to be careful with terminology here.  The authors state: “The outcome variable of the model was the concentration of each metabolite, while the major predictor variable of interest was the beta estimate of year, which represents the overall changes in metabolite concentration each year.” It is true that in this longitudinal study, significant effects of year would capture the effects of ageing on metabolite levels. However, when we talk about “biological age” versus “chronological age” in the literature, we typically are referring to the underlying health status of an individual, which is determined in part by his or her chronological age, and in part by residual variation that is not explained by age. For example, an individual that is 60 years old has a particular life expectancy based on age alone. But if that individual’s _biological age_ is older than that individual’s chronological age, we would expect a shorter life expectancy.

  Perhaps this is just a matter of terminology. I would agree with the claim that this analysis shows age-related change in metabolite levels. But as currently carried out, these results do not point to changes affected by biological age, nor is it clear how biological age would be measured here. I would encourage the authors to consider studies such as those on the methylome clock by Dr. Steve Horvath for an example of the distinction between chronological and biological age.

 In fact, by controlling for various traits known to be associated with aging (smoking status, physical activity, blood pressure), the authors might actually limit the potential to determine whether longitudinal changes in metabolite levels are predictive of these factors.

 Given the nature of the available data, I actually think the authors might be able to carry out an analysis that looks at the degree to which metabolites are associated with biological age. However, as it currently stands, I don’t think they have yet done so.

 If the intention was simply to show that there are clear age-related trends, even after controlling for chronological age and other factors, the authors have done so, but need to reframe how they present the evidence.

Finally, I would have liked to see a multivariate analysis of the data, in addition to the GEE approach employed here, to place this work more in the context of other studies on age-related variation in the metabolome.

Minor points:

There are a few grammatical errors throughout the manuscript. The entire manuscript should be checked thoroughly. For example:

L60: biomarkers remains a great challenge at technical level

L78: Consequently, previous studies have taken metabolomics approach

L86: to track the changes in metabolite concentration of each individual along time

L345. “CV” ≠ “coefficient of value”

The authors should note that these studies necessarily measure steady state levels of metabolites, rather than flux through pathways.

Author Response

Response to Reviewer 2 Comments

This manuscript presents results from a longitudinal study of the targeted metabolome at two time points in a human longitudinal study cohort, with replication carried out in a second cohort. The authors use Generalized Estimating Equations to control for a variety of health-related traits (e.g., smoking, physical activity, blood pressure, chronological age), and then test for significant effects of time on the level of individual metabolites.

Over half the metabolites in the first (KORA) cohort were associated with what the authors refer to as “biological ageing”, though only a handful of these metabolites were replicated in the second study.

The authors claim that this the first longitudinal study of metabolites in humans. Previous work has already been done in this regard (e.g. Mielke et al. 2014 Aging Cell; Saleem et al. In Press. Journals of Gerontology), though there are indeed relatively few studies, and as such, this work would provide further valuable data regarding our understanding of how metabolite levels change with age.

Response 1: We thank this reviewer for his willingness to participate in the peer-review processing of our manuscript and her/his constructive comments to improve our study.

We have revised our manuscript and corrected this claim (that this the first longitudinal study of metabolites in humans) by replacing the term ‘absent’ with ‘limited’, as shown in lines 86-87: ‘However, to our knowledge, such a population-based longitudinal analysis is by far limited in targeted metabolomics studies regarding to human ageing [6]’. Additionally, as Mielke et al. 2014 Aging Cell study investigated plasma ceramides and dihydroceramides, use a different panel of metabolites.

Although the data are valuable, I have some concerns regarding the interpretation of these results.  The authors claim that this study measures the effect of biological ageing rather than chronological ageing. One needs to be careful with terminology here.  The authors state: “The outcome variable of the model was the concentration of each metabolite, while the major predictor variable of interest was the beta estimate of year, which represents the overall changes in metabolite concentration each year.” It is true that in this longitudinal study, significant effects of year would capture the effects of ageing on metabolite levels. However, when we talk about “biological age” versus “chronological age” in the literature, we typically are referring to the underlying health status of an individual, which is determined in part by his or her chronological age, and in part by residual variation that is not explained by age. For example, an individual that is 60 years old has a particular life expectancy based on age alone. But if that individual’s _biological age_ is older than that individual’s chronological age, we would expect a shorter life expectancy.

Perhaps this is just a matter of terminology. I would agree with the claim that this analysis shows age-related change in metabolite levels. But as currently carried out, these results do not point to changes affected by biological age, nor is it clear how biological age would be measured here. I would encourage the authors to consider studies such as those on the methylome clock by Dr. Steve Horvath for an example of the distinction between chronological and biological age.

Response 2: We agree to the reviewer that this is a problem of terminology and that our label ‘biological ageing’ promotes misunderstandings. To further avoid such ambiguities we replaced the term ‘biological ageing’ to the reduced term ‘ageing’ through the whole manuscript.

 In fact, by controlling for various traits known to be associated with aging (smoking status, physical activity, blood pressure), the authors might actually limit the potential to determine whether longitudinal changes in metabolite levels are predictive of these factors.

Response 3: We agree to the reviewer, that there might be age-related metabolites we miss to detect by adjusting our models with a number of confounder. On the other side, by adjusting for fewer confounders, we could not guarantee anymore, that our findings are specifically associated with ageing. We prefer to report findings we are perfectly convinced.

Furthermore, the aim of our current study is not “whether longitudinal changes in metabolite levels are predictive of these factors (“traits known to be associated with aging”). Those traits are not only associated with ageing, but also with the metabolites, and are thus confounders. In order to interpret the association between longitudinal changes in metabolite levels and age we need to control for the confounders.

Given the nature of the available data, I actually think the authors might be able to carry out an analysis that looks at the degree to which metabolites are associated with biological age. However, as it currently stands, I don’t think they have yet done so. If the intention was simply to show that there are clear age-related trends, even after controlling for chronological age and other factors, the authors have done so, but need to reframe how they present the evidence.

Response 4: In fact, the intention of this study was to show that there are age related trends and name them. Please see more in Response 2.

Finally, I would have liked to see a multivariate analysis of the data, in addition to the GEE approach employed here, to place this work more in the context of other studies on age-related variation in the metabolome.

Response 5: To address this concern, we used the women-dataset to additionally calculate a Linear Regression Model considering the combination of all significant metabolites. The following results were obtained:

               Estimate  Std. Error     t value      p  valueorn          2.20389257 0.208928183  10.5485653  5.157868e-26arg         -2.54459869 0.171529069 -14.8347956  8.726928e-50ser          0.10958399 0.210013124   0.5217959  6.018124e-01tyr          1.30590597 0.188932322   6.9120305  4.777652e-12c18          0.05423404 0.154616761   0.3507643  7.257652e-01

Serine and c18 are not significant anymore. With respect to Serine we observed high anti-correlations with Ornithine (r =  -0.43) and Arginine (r =  -0.58). The liner combination of these metabolites is therefore highly likely to substitute Serine and therefore make it less relevant in this calculation. However, such observations are expected. Also in further, similar analyses, we did not observe any additional results that would change anything about our conclusion.

Minor points:

There are a few grammatical errors throughout the manuscript. The entire manuscript should be checked thoroughly. For example:

L60: biomarkers remains a great challenge at technical level

L78: Consequently, previous studies have taken metabolomics approach

L86: to track the changes in metabolite concentration of each individual along time

L345. “CV” ≠ “coefficient of value”

Response 6: We thank the reviewer for the hints. All points were corrected.

The authors should note that these studies necessarily measure steady state levels of metabolites, rather than flux through pathways.

Response 7: We thank the reviewer for the hint.

Reviewer 3 Report

Review of MDPI Article by Chak et al ‘Ageing investigation using two-time-point metabolomics data from KORA and CARLA studies’

Summary

This is an interesting paper describing the elucidation of human serum metabolites that are associated with biological ageing as opposed to chronological ageing in the KORA and CARLA cohorts

Significant Comments

1.     Introduction lines 70 to 77: the singular focus on targeted metabolic profiling is inappropriate: targeted methods will only measure selected metabolites, by definition. Given the complexity of human biology and ageing and our incomplete knowledge and understanding of these topics, it is illogical to take solely a targeted approach. An untargeted metabolic profiling approach would be preferable to a targeted approach, or at least should have been conducted alongside the targeted methods? The metabolites discovered by the authors might well be less relevant than thise that could have been discovered with an untargeted approach.

2.     Introduction: Study Design. Given the significant variability in human serum metabolite profiles it is inappropriate to study just two timepoints. The changes seen in metabolite profiles could be due to some variation, other than ageing: a multi-timepoint study design would have helped confirm the association of particular metabolites with biological ageing as opposed to other factors. This is mentioned at the end of the paper.

3.     lines 145 to 148: in addition to 5 - and 4 - pFDR significantly changed metabolites found in women and men between the two studies, another 5 metabolites displayed opposite trends between the discovery and replication cohorts, indicating that there is no general association between the levels of the latter 5 metabolites and biological ageing.  The authors do discuss why this could occur later in the manuscript but this is a significant concern in terms of whether any of the other metabolites discovered would be replicated in future studies. The text on lines 145 to 148 strangely just mentions phenylalanine but the identities of the other 4 metabolites are not given at this point.

4.     the tables of metabolites are difficult to understand. Table S1: ‘The list of 123 metabolites measured and passed quality control in KORA S4 and F4.’  conflicts with Table S2:  Changes in metabolite concentration in KORA Study in women (N = 317) and men (N = 273) over seven-year period. The latter table lists the following metabolites: Sum of hexoses, Arginine, Glutamine, Glycine, Histidine, Methionine, Ornithine, Phenylalanine, Proline, Serine, Threonine Tryptophan, Tyrosine, Valine, xLeucine at the end. These metabolites are not listed in Table S1 and therefore it is difficult to understand why they are in Table S2. Line 333 indicates these metabolites should be part of the 123. It is unclear what xLeucine refers to: presumably, this is either leucine or isoleucine? The Table is unclear.

5.     lines 263-264. The comments: ‘Moreover, both KORA and CARLA studies made use of serum samples, which are generally more stable over urine’ is not really appropriate. Serum is much more subject to homeostatic control than urine and is therefore generally less sensitive to intervention-created metabolic phenotype changes. Differences in urinary dilution due to differences in water intake are readily corrected with normalization procedures. It would be important and interesting to repeat these studies with urine samples.

Detailed Comments

1.     line 63: the phrase ‘end products of metabolism that highly involve in biochemical pathways,’ makes no sense

2.     line 64 ditto: this says ‘not only they are the most’ but presumably should say ‘not only are they the most’

3.     line 77 – the use of uprising is incorrect: upcoming?

Author Response

Response to Reviewer 3 Comments

Summary

This is an interesting paper describing the elucidation of human serum metabolites that are associated with biological ageing as opposed to chronological ageing in the KORA and CARLA cohorts

We thank this reviewer for his willingness to participate in the peer-review processing of our manuscript and her/his constructive comments to improve our study.

Significant Comments

1.     Introduction lines 70 to 77: the singular focus on targeted metabolic profiling is inappropriate: targeted methods will only measure selected metabolites, by definition. Given the complexity of human biology and ageing and our incomplete knowledge and understanding of these topics, it is illogical to take solely a targeted approach. An untargeted metabolic profiling approach would be preferable to a targeted approach, or at least should have been conducted alongside the targeted methods? The metabolites discovered by the authors might well be less relevant than thise that could have been discovered with an untargeted approach.

Response 1: We agree with the use of untargeted approach could help us to identify more potential metabolites related to ageing. Our idea to focus on targeted metabolomics is because we would like to particularly focus on the metabolites, which are biochemically known and annotated so that we could also better understand the potential biochemical pathways of the identified ageing-associated metabolites.

2.     Introduction: Study Design. Given the significant variability in human serum metabolite profiles it is inappropriate to study just two time points. The changes seen in metabolite profiles could be due to some variation, other than ageing: a multi-time point study design would have helped confirm the association of particular metabolites with biological ageing as opposed to other factors. This is mentioned at the end of the paper.

Response 2: We are aware of our limitation of having two time-point data only and have addressed it. We look forward to seeing studies that present multi-time-points data in the future.

3.     lines 145 to 148: in addition to 5 - and 4 - pFDR significantly changed metabolites found in women and men between the two studies, another 5 metabolites displayed opposite trends between the discovery and replication cohorts, indicating that there is no general association between the levels of the latter 5 metabolites and biological ageing.  The authors do discuss why this could occur later in the manuscript but this is a significant concern in terms of whether any of the other metabolites discovered would be replicated in future studies. The text on lines 145 to 148 strangely just mentions phenylalanine but the identities of the other 4 metabolites are not given at this point.

Response 3: We have updated the identities of the other five metabolites that were in opposite changes. As shown in the revised manuscript: ‘In addition to the replicated metabolites mentioned above, additional five metabolites  (namely octadecenoylcarnitine (C18:1), phenylalanine, valine, lysoPC a C16:0 and lysoPC a C18:0)’.

4.     the tables of metabolites are difficult to understand. Table S1: ‘The list of 123 metabolites measured and passed quality control in KORA S4 and F4.’  conflicts with Table S2:  Changes in metabolite concentration in KORA Study in women (N = 317) and men (N = 273) over seven-year period. The latter table lists the following metabolites: Sum of hexoses, Arginine, Glutamine, Glycine, Histidine, Methionine, Ornithine, Phenylalanine, Proline, Serine, Threonine Tryptophan, Tyrosine, Valine, xLeucine at the end. These metabolites are not listed in Table S1 and therefore it is difficult to understand why they are in Table S2. Line 333 indicates these metabolites should be part of the 123. It is unclear what xLeucine refers to: presumably, this is either leucine or isoleucine? The Table is unclear.

Response 4: Thank you very much for pointing out the problem of Supplementary table S1. We would like to apologize for the confusion caused. The original table contains 123 metabolites, but part of the metabolites (e.g. H1 and amino acids) was accidentally missed out during editing the content. We have updated the Supplementary table S1 accordingly. xLeucine is the sum of leucine and isoleucine. The information is also updated in the legends of supplementary tables.

5.     lines 263-264. The comments: ‘Moreover, both KORA and CARLA studies made use of serum samples, which are generally more stable over urine’ is not really appropriate. Serum is much more subject to homeostatic control than urine and is therefore generally less sensitive to intervention-created metabolic phenotype changes. Differences in urinary dilution due to differences in water intake are readily corrected with normalization procedures. It would be important and interesting to repeat these studies with urine samples.

Response 5: We specified the statement in this revised manuscript as the following: ‘Moreover, both KORA and CARLA studies made use of serum samples, which are generally more subject to homeostatic control and hence generally less sensitive to intervention-induced metabolic phenotype changes compared to urine samples.’

Detailed Comments

1.     line 63: the phrase ‘end products of metabolism that highly involve in biochemical pathways,’ makes no sense

2.     line 64 ditto: this says ‘not only they are the most’ but presumably should say ‘not only are they the most’

3.     line 77 – the use of uprising is incorrect: upcoming?

Response 6: We thank for the hints and corrected all errors.

Round  2

Reviewer 1 Report

The authors have addressed relevant comments from the first round of review. They have moderated the tone of several statements to avoid making overly strong claims and have included new references.

Some minor issues remain:

For example, the conclusion basically repeats previous statements in the introduction. The authors can avoid this using specific information. It would be better to include a few selected biologically relevant metabolites and their pathway associations, just as the authors did in the abstract. Specificity is almost always better than vagueness.

Author Response

Response to Reviewer 1 Round 2 Comments

The authors have addressed relevant comments from the first round of review. They have moderated the tone of several statements to avoid making overly strong claims and have included new references.

Some minor issues remain:

For example, the conclusion basically repeats previous statements in the introduction. The authors can avoid this using specific information. It would be better to include a few selected biologically relevant metabolites and their pathway associations, just as the authors did in the abstract. Specificity is almost always better than vagueness.

Response: We thank the reviewer for his/her willingness to participate in the second round of peer-review processing of our manuscript and the constructive comments to improve our study.

The conclusion in the manuscript was updated accordingly. We have included the three metabolites (arginine, ornithine and serine) and their associations with ageing process in the conclusion part as suggested by the reviewer.

Reviewer 2 Report

The authors have carefully considered the comments from a previous round of reviews and responded accordingly. I think the clear strength of this work is that it provides a longitudinal analysis of age-related changes in metabolite concentrations from a targeted profile, that it does so in both men and women, and that the results are replicated in two independent cohorts.

These data are clearly worth publishing. However, I do have some remaining concerns with the presentation and interpretation.

First, the authors claim to have identified “ageing” associated changes in metabolites. The power of the longitudinal analysis is that it identifies age associated changes after controlling for any cohort effects (different years of birth, different environment, etc.). However, there are no measures here of the ageing process per se, and so one cannot know if the observed “age associated” changes are also “ageing associated”. For example, if it was observed that age-related changes in some metabolite were associated with increased measures of inflammation, or increased incidence of an age-related disease, one could make a stronger argument that these age-associated changes are actually associated with the process of ageing. The authors are rightly excited about the potential for metabolite levels to serve as biomarkers of ageing. However, this is very different from serving as biomarkers of age. I would argue that the present data are at the stage of clearly having potential as biomarkers of age, but not of ageing. To summarize, it is as if the authors define ageing as “the change from age x to age x+delta x”. This is ageing in a chronological sense, but not necessarily in a biological sense.

I was very interested in the sex-specific differences in the data. I would have liked to see explicit statistical tests for differences. If a metabolite increases significantly with age in one sex but not the other, that alone does not allow us to say the sexes are significantly different. Rather, one would need to incorporate additional terms in the GEE model—sex, and sex-x-age interaction. The interaction term would provide an explicit statistical test for differences between males and females.

The authors note that several metabolites (5 in men, 4 in women) overlapped between the two cohorts. Was this either significantly more or less than expected by chance?

Figure 2 shows age- and sex-specific changes in the distribution of metabolite levels for several features. I would have liked to see an illustration that showed the variation in a longitudinal fashion. These figures do not capture to valuable realism of the GEE model.

Minor points:

The first sentence claims that ageing is “the process of progressive physiological losses in energy homeostasis efficience” due to “accumulation of cellular damage”. This is a hypothesis, rather than a definition.

Would it be possible to test for effects of baseline age on the degree of change in metabolites, to test for non-linear changes with age?

Author Response

Response to Reviewer 2 Round 2 Comments

The authors have carefully considered the comments from a previous round of reviews and responded accordingly. I think the clear strength of this work is that it provides a longitudinal analysis of age-related changes in metabolite concentrations from a targeted profile, that it does so in both men and women, and that the results are replicated in two independent cohorts.

Response 1: We thank the reviewer for his/her willingness to participate in the second round of peer-review processing of our manuscript and the constructive comments to improve our study.

We would like to thank the reviewer for highlighting the strengths of the work. We have updated the text in the section of strengths and limitations.

These data are clearly worth publishing. However, I do have some remaining concerns with the presentation and interpretation.

First, the authors claim to have identified “ageing” associated changes in metabolites. The power of the longitudinal analysis is that it identifies age associated changes after controlling for any cohort effects (different years of birth, different environment, etc.). However, there are no measures here of the ageing process per se, and so one cannot know if the observed “age associated” changes are also “ageing associated”. For example, if it was observed that age-related changes in some metabolite were associated with increased measures of inflammation, or increased incidence of an age-related disease, one could make a stronger argument that these age-associated changes are actually associated with the process of ageing. The authors are rightly excited about the potential for metabolite levels to serve as biomarkers of ageing. However, this is very different from serving as biomarkers of age. I would argue that the present data are at the stage of clearly having potential as biomarkers of age, but not of ageing. To summarize, it is as if the authors define ageing as “the change from age x to age x+delta x”. This is ageing in a chronological sense, but not necessarily in a biological sense.

Response 2: When we consider the term “ageing” in our manuscript, it refers to both chronological ageing for all individuals (in terms of passing of time) but also potentially biological ageing process (which we believe, metabolite changes give us some insights to this process). We agree that ageing refer in our manuscript do not necessarily in a biological sense, but this is exactly the question that we would like to raise. It is true that the power of longitudinal analysis shows us metabolite changes over time (annually in chronological sense) after controlling for baseline age and other factors. However, the reason why we would like to control for baseline age is because we can see if the metabolite changes are significantly different among individuals of different age. When there is a significant trend in metabolite changes when all individuals age chronologically in terms of year, it also gives us information on the “outliners”. Do these outliner individuals tell us something about different biological ageing rate based on these metabolites?

We disagree with the reviewer that the finding of this study is “age-associated” change because we cancelled out the effect of baseline age so the actual age of individuals is not our main focus. What we are trying to see is the “ageing-associated” changes in metabolites over time, when everyone experiences the same chronological ageing.

I was very interested in the sex-specific differences in the data. I would have liked to see explicit statistical tests for differences. If a metabolite increases significantly with age in one sex but not the other, that alone does not allow us to say the sexes are significantly different. Rather, one would need to incorporate additional terms in the GEE model—sex, and sex-x-age interaction. The interaction term would provide an explicit statistical test for differences between males and females.

Response 3: We performed a new calculation, the effect of sex-x-age interaction term for each metabolite is shown below:

                      beta betavl betavr             betacl     p-value      pFDRc0             -0.46889965  -0.62  -0.32 -0.47(-0.62,-0.32) 2.304264e-09 6.856591e-09c2             -0.02110534  -0.19   0.14  -0.02(-0.19,0.14) 8.016975e-01 8.083231e-01c3             -0.46515749  -0.62  -0.31 -0.47(-0.62,-0.31) 3.612927e-09 1.001766e-08c4             -0.18226452  -0.34  -0.02 -0.18(-0.34,-0.02) 2.779632e-02 3.569633e-02c4_1_dc__c6    -0.10552357  -0.26   0.05  -0.11(-0.26,0.05) 1.922562e-01 2.151858e-01c5             -0.57954978  -0.73  -0.43 -0.58(-0.73,-0.43) 2.357605e-14 1.065288e-13c8             -0.18147915  -0.34  -0.02 -0.18(-0.34,-0.02) 2.729765e-02 3.542887e-02c10            -0.23113096  -0.40  -0.06  -0.23(-0.4,-0.06) 7.483583e-03 1.049862e-02c10_1          -0.27546817  -0.44  -0.11 -0.28(-0.44,-0.11) 1.298560e-03 2.112325e-03c10_2          -0.26908839  -0.44  -0.10  -0.27(-0.44,-0.1) 1.515143e-03 2.432204e-03c12            -0.31254115  -0.48  -0.14 -0.31(-0.48,-0.14) 2.665548e-04 4.580237e-04c14            -0.17972380  -0.35  -0.01 -0.18(-0.35,-0.01) 3.657544e-02 4.553269e-02c14_1          -0.12088852  -0.28   0.04  -0.12(-0.28,0.04) 1.467192e-01 1.657383e-01c14_2          -0.32191807  -0.49  -0.15 -0.32(-0.49,-0.15) 2.180515e-04 3.855404e-04c16            -0.44342030  -0.60  -0.29  -0.44(-0.6,-0.29) 1.962366e-08 4.885891e-08c16_1          -0.02339839  -0.19   0.15  -0.02(-0.19,0.15) 7.864028e-01 8.062281e-01c18            -0.50231617  -0.66  -0.34  -0.5(-0.66,-0.34) 9.234011e-10 3.044728e-09c18_1          -0.35614430  -0.52  -0.19 -0.36(-0.52,-0.19) 1.800484e-05 3.542888e-05c18_2          -0.50009595  -0.67  -0.33  -0.5(-0.67,-0.33) 4.000572e-09 1.084599e-08pc_aa_c28_1     0.67292714   0.53   0.82    0.67(0.53,0.82) 3.883057e-20 2.786665e-19pc_aa_c30_0     0.40687789   0.26   0.55    0.41(0.26,0.55) 4.140298e-08 1.010233e-07pc_aa_c32_0     0.14397071  -0.01   0.29   0.14(-0.01,0.29) 5.877420e-02 7.242882e-02pc_aa_c32_1     0.32934640   0.18   0.48    0.33(0.18,0.48) 2.051454e-05 3.923621e-05pc_aa_c32_2     0.58422135   0.44   0.73    0.58(0.44,0.73) 4.608498e-16 2.555622e-15pc_aa_c32_3     0.92847078   0.79   1.07    0.93(0.79,1.07) 4.797359e-37 1.950926e-35pc_aa_c34_1     0.25024566   0.10   0.40      0.25(0.1,0.4) 9.030704e-04 1.509241e-03pc_aa_c34_2     0.32340131   0.18   0.47    0.32(0.18,0.47) 8.858708e-06 1.801271e-05pc_aa_c34_3     0.68626088   0.54   0.84    0.69(0.54,0.84) 1.688124e-19 1.144173e-18pc_aa_c34_4     0.59553224   0.44   0.75     0.6(0.44,0.75) 1.999764e-14 9.383506e-14pc_aa_c36_0     0.06949103  -0.06   0.20    0.07(-0.06,0.2) 2.933826e-01 3.224565e-01pc_aa_c36_1     0.35511694   0.20   0.51     0.36(0.2,0.51) 6.009679e-06 1.264105e-05pc_aa_c36_2     0.36314532   0.23   0.50     0.36(0.23,0.5) 1.270107e-07 2.923643e-07pc_aa_c36_3     0.54619698   0.40   0.69     0.55(0.4,0.69) 6.257726e-13 2.385758e-12pc_aa_c36_4     0.33195172   0.18   0.49    0.33(0.18,0.49) 2.956089e-05 5.548353e-05pc_aa_c36_5     0.12800993  -0.03   0.29   0.13(-0.03,0.29) 1.134095e-01 1.305279e-01pc_aa_c36_6     0.49138478   0.34   0.65    0.49(0.34,0.65) 3.687820e-10 1.285469e-09pc_aa_c38_0     0.16948147   0.02   0.32    0.17(0.02,0.32) 2.853580e-02 3.626425e-02pc_aa_c38_3     0.59262993   0.45   0.74    0.59(0.45,0.74) 1.684676e-15 8.936106e-15pc_aa_c38_4     0.39061263   0.24   0.54    0.39(0.24,0.54) 6.299643e-07 1.423253e-06pc_aa_c38_5     0.32940990   0.17   0.48    0.33(0.17,0.48) 3.196673e-05 5.909002e-05pc_aa_c38_6     0.21419035   0.06   0.37    0.21(0.06,0.37) 8.291901e-03 1.149559e-02pc_aa_c40_2     0.12757902  -0.02   0.27   0.13(-0.02,0.27) 8.858723e-02 1.049286e-01pc_aa_c40_3     0.23545321   0.09   0.38    0.24(0.09,0.38) 1.766273e-03 2.798510e-03pc_aa_c40_4     0.22846723   0.07   0.38    0.23(0.07,0.38) 3.941673e-03 5.864440e-03pc_aa_c40_5     0.19962773   0.05   0.35     0.2(0.05,0.35) 1.135068e-02 1.538648e-02pc_aa_c40_6     0.20030838   0.05   0.36     0.2(0.05,0.36) 1.126403e-02 1.538648e-02pc_aa_c42_0     0.19377956   0.04   0.35    0.19(0.04,0.35) 1.207904e-02 1.619388e-02pc_aa_c42_1     0.13687756  -0.02   0.29   0.14(-0.02,0.29) 8.391029e-02 1.003633e-01pc_aa_c42_2     0.01980008  -0.13   0.17   0.02(-0.13,0.17) 7.951446e-01 8.083231e-01pc_aa_c42_4     0.19339004   0.05   0.33    0.19(0.05,0.33) 7.145341e-03 1.025567e-02pc_aa_c42_5     0.26314227   0.11   0.42    0.26(0.11,0.42) 1.034355e-03 1.705287e-03pc_aa_c42_6     0.36378206   0.21   0.52    0.36(0.21,0.52) 3.231470e-06 6.916479e-06pc_ae_c30_0     0.54746406   0.40   0.69     0.55(0.4,0.69) 5.449290e-14 2.374334e-13pc_ae_c32_1     0.44473661   0.30   0.59     0.44(0.3,0.59) 1.457618e-09 4.559727e-09pc_ae_c32_2     0.76213889   0.63   0.90     0.76(0.63,0.9) 8.262442e-29 1.260022e-27pc_ae_c34_0     0.49793439   0.35   0.65     0.5(0.35,0.65) 6.420486e-11 2.373634e-10pc_ae_c34_1     0.70286050   0.56   0.84     0.7(0.56,0.84) 2.416712e-22 1.965592e-21pc_ae_c34_2     0.66304316   0.51   0.81    0.66(0.51,0.81) 4.589561e-18 2.799632e-17pc_ae_c34_3     0.45570639   0.31   0.61    0.46(0.31,0.61) 2.455564e-09 7.132830e-09pc_ae_c36_1     0.71014563   0.57   0.85    0.71(0.57,0.85) 5.487726e-24 5.579188e-23pc_ae_c36_2     0.72420068   0.58   0.87    0.72(0.58,0.87) 8.063219e-24 7.567021e-23pc_ae_c36_3     0.58922219   0.44   0.74    0.59(0.44,0.74) 6.540895e-14 2.751687e-13pc_ae_c36_4     0.13369995  -0.02   0.29   0.13(-0.02,0.29) 9.630901e-02 1.129779e-01pc_ae_c36_5     0.06972620  -0.09   0.22   0.07(-0.09,0.22) 3.777798e-01 4.078684e-01pc_ae_c38_0     0.46038622   0.31   0.61    0.46(0.31,0.61) 1.859760e-09 5.672267e-09pc_ae_c38_1     0.19106099   0.03   0.35    0.19(0.03,0.35) 1.759177e-02 2.332822e-02pc_ae_c38_2     0.55869150   0.42   0.70     0.56(0.42,0.7) 6.942904e-15 3.529309e-14pc_ae_c38_3     0.84915951   0.71   0.99    0.85(0.71,0.99) 1.294917e-32 3.159598e-31pc_ae_c38_4     0.46475232   0.32   0.61    0.46(0.32,0.61) 3.564190e-10 1.278915e-09pc_ae_c38_5     0.06653568  -0.08   0.22   0.07(-0.08,0.22) 3.872750e-01 4.144522e-01pc_ae_c38_6     0.23884912   0.09   0.39    0.24(0.09,0.39) 2.233155e-03 3.448669e-03pc_ae_c40_1     0.21047005   0.06   0.36    0.21(0.06,0.36) 7.486722e-03 1.049862e-02pc_ae_c40_2     0.45007326   0.30   0.60      0.45(0.3,0.6) 2.644171e-09 7.502067e-09pc_ae_c40_3     0.76082514   0.63   0.89    0.76(0.63,0.89) 6.907876e-31 1.203944e-29pc_ae_c40_4     0.44167218   0.30   0.58     0.44(0.3,0.58) 1.070478e-09 3.436798e-09pc_ae_c40_5     0.26416599   0.13   0.40     0.26(0.13,0.4) 1.226035e-04 2.232481e-04pc_ae_c40_6     0.32051498   0.18   0.46    0.32(0.18,0.46) 7.026454e-06 1.452928e-05pc_ae_c42_1     0.25428575   0.11   0.40     0.25(0.11,0.4) 5.944241e-04 1.007219e-03pc_ae_c42_2     0.35451072   0.21   0.50     0.35(0.21,0.5) 2.556356e-06 5.594877e-06pc_ae_c42_3     0.31501244   0.17   0.46    0.32(0.17,0.46) 2.058293e-05 3.923621e-05pc_ae_c42_4     0.29105275   0.14   0.44    0.29(0.14,0.44) 1.971720e-04 3.537498e-04pc_ae_c42_5     0.18191461   0.03   0.33    0.18(0.03,0.33) 1.801869e-02 2.363743e-02pc_ae_c44_3     0.14456507  -0.02   0.31   0.14(-0.02,0.31) 7.988912e-02 9.746473e-02pc_ae_c44_4     0.23384217   0.08   0.39    0.23(0.08,0.39) 3.007107e-03 4.585837e-03pc_ae_c44_5     0.02271000  -0.13   0.17   0.02(-0.13,0.17) 7.667051e-01 7.994703e-01pc_ae_c44_6     0.07484224  -0.08   0.23   0.07(-0.08,0.23) 3.447683e-01 3.755511e-01lysopc_a_c16_0 -0.36357494  -0.50  -0.23  -0.36(-0.5,-0.23) 1.243196e-07 2.916729e-07lysopc_a_c16_1 -0.07666071  -0.22   0.07  -0.08(-0.22,0.07) 2.896474e-01 3.212453e-01lysopc_a_c17_0  0.11596960  -0.03   0.26   0.12(-0.03,0.26) 1.154257e-01 1.316069e-01lysopc_a_c18_0 -0.23037074  -0.38  -0.08 -0.23(-0.38,-0.08) 1.898960e-03 2.970168e-03lysopc_a_c18_1 -0.54453533  -0.69  -0.40  -0.54(-0.69,-0.4) 1.616936e-13 6.363427e-13lysopc_a_c18_2 -0.71907620  -0.87  -0.57 -0.72(-0.87,-0.57) 1.982323e-21 1.511521e-20lysopc_a_c20_3 -0.38528820  -0.55  -0.22 -0.39(-0.55,-0.22) 2.568140e-06 5.594877e-06lysopc_a_c20_4 -0.60081539  -0.76  -0.44  -0.6(-0.76,-0.44) 1.607356e-13 6.363427e-13sm__oh__c14_1   0.69578315   0.56   0.83     0.7(0.56,0.83) 2.645886e-24 2.934528e-23sm__oh__c16_1   0.69228052   0.56   0.83    0.69(0.56,0.83) 1.341939e-23 1.169404e-22sm__oh__c22_1   0.65614815   0.53   0.78    0.66(0.53,0.78) 4.238618e-25 5.171114e-24sm__oh__c22_2   0.95071062   0.83   1.07    0.95(0.83,1.07) 5.894092e-54 7.190792e-52sm__oh__c24_1   0.37873463   0.25   0.51    0.38(0.25,0.51) 1.570017e-08 3.990460e-08sm_c16_0        0.40574746   0.26   0.55    0.41(0.26,0.55) 4.623769e-08 1.106078e-07sm_c16_1        0.83444220   0.71   0.96    0.83(0.71,0.96) 2.969992e-37 1.811695e-35sm_c18_0        0.64576396   0.50   0.79     0.65(0.5,0.79) 1.026630e-17 5.964231e-17sm_c18_1        0.87063120   0.74   1.01    0.87(0.74,1.01) 1.192078e-36 3.635839e-35sm_c20_2        0.64074994   0.52   0.76    0.64(0.52,0.76) 2.073317e-25 2.810496e-24sm_c24_0        0.14818947   0.01   0.28    0.15(0.01,0.28) 2.967388e-02 3.732179e-02sm_c24_1        0.19405894   0.05   0.33    0.19(0.05,0.33) 6.912444e-03 1.003950e-02sm_c26_1        0.02092384  -0.13   0.17   0.02(-0.13,0.17) 7.856820e-01 8.062281e-01h1             -0.20317235  -0.34  -0.07  -0.2(-0.34,-0.07) 3.644487e-03 5.489228e-03arg             0.01176168  -0.14   0.16   0.01(-0.14,0.16) 8.762413e-01 8.762413e-01gln            -0.14399207  -0.32   0.03  -0.14(-0.32,0.03) 1.075039e-01 1.249093e-01gly             0.45854902   0.30   0.62     0.46(0.3,0.62) 1.307691e-08 3.394432e-08his             0.04407419  -0.11   0.20    0.04(-0.11,0.2) 5.702217e-01 6.049308e-01met            -0.50823129  -0.67  -0.35 -0.51(-0.67,-0.35) 5.863546e-10 1.987091e-09orn            -0.12272238  -0.26   0.02  -0.12(-0.26,0.02) 8.321497e-02 1.003633e-01phe            -0.36821672  -0.53  -0.20  -0.37(-0.53,-0.2) 1.089942e-05 2.179885e-05pro            -0.63683917  -0.78  -0.49 -0.64(-0.78,-0.49) 4.164316e-18 2.673929e-17ser             0.23396201   0.07   0.40     0.23(0.07,0.4) 6.087764e-03 8.948280e-03thr            -0.03289619  -0.20   0.14   -0.03(-0.2,0.14) 7.072878e-01 7.438716e-01trp            -0.53405400  -0.72  -0.35 -0.53(-0.72,-0.35) 7.888930e-09 2.092282e-08tyr            -0.29666829  -0.46  -0.14  -0.3(-0.46,-0.14) 2.458996e-04 4.285679e-04val            -0.62225180  -0.78  -0.46 -0.62(-0.78,-0.46) 1.064887e-14 5.196648e-14xleu_S4          -0.65266339  -0.76  -0.54 -0.65(-0.76,-0.54) 2.243494e-31 4.561772e-30

The authors note that several metabolites (5 in men, 4 in women) overlapped between the two cohorts. Was this either significantly more or less than expected by chance?

Response 4: We did not only observe the significance but also the comparable degree of variation of metabolites (beta estimates) over time. We have also tried to minimize the bias resulted from the differences in KORA and CARLA cohort study to increase comparability of the cohort data. We have reasons to believe that the metabolites are not significant by chance.

Figure 2 shows age- and sex-specific changes in the distribution of metabolite levels for several features. I would have liked to see an illustration that showed the variation in a longitudinal fashion. These figures do not capture to valuable realism of the GEE model.

Response 5: We have attempted to capture the variation in longitudinal fashion visually via residual plots and spaghetti plots, however we think that would be redundant to the data we present (in Table 2), also the data speak for itself. Hence we decided to use the current distribution of metabolites to capture the fundamental difference in metabolite distribution without controlling for individual factors to add information to the interpretation.              

Minor points:

The first sentence claims that ageing is “the process of progressive physiological losses in energy homeostasis efficience” due to “accumulation of cellular damage”. This is a hypothesis, rather than a definition.

Response 6: We thank for the comment and have updated the sentences and updated the citations.

Would it be possible to test for effects of baseline age on the degree of change in metabolites, to test for non-linear changes with age?

Response 7: The test was carried out before and the non-linear effect was not significant.
